# Board Attributes and Bank Performance in Light of Saudi Corporate Governance Regulations

Omer Saeed Habtoor [1,2]

1   Department of Administrative Sciences, Applied College, Northern Border University, Rafha 76321, Saudi Arabia; omer.habtoor@nbu.edu.sa or omerhabtoor@hotmail.com; Tel.: +966-552667691
2   Department of Accounting, Faculty of Administrative Sciences, Aden University, Aden P.O. Box 6312, Yemen

**Abstract:** This study investigates the relationship between various attributes of boards of directors on bank performance in light of Saudi corporate governance regulations. The data set of this study is extracted from the annual reports of all 12 banks listed on the Saudi Stock Exchange (Tadawul) over a period of 10 years from 2009 to 2018. To test the study hypotheses, check the robustness of the results, and address potential endogeneity issues, this study applies different statistical methods, including FGLS, OLS, RE, PLCSE, and 2SLS, using STATA version 17. The results of multivariate analysis show that board size has a significant positive influence only on operational bank performance (ROA). For board composition, the results show that while board independence has a significant negative impact on accounting-based performance (ROA and ROE), it affects positively and significantly the market-based performance (Tobin's Q). Regarding board education, the results indicate that board members with at least a Bachelor's degree have a significant negative impact on ROA and ROE. In contrast, PhD holders on the board have a significant positive impact on ROA and ROE, while Master's holders affect positively and significantly all measures of bank performance. With respect to board diversity, only the CEO nationality has a significant positive effect on ROA and ROE. Board IT experience is found to be significantly and positively associated with ROA and ROE, while board meeting attendance has a significant positive influence only on ROE. These findings have important implications, especially for Saudi regulatory authorities to assess the current practice and compliance with the Saudi corporate governance regulations (SCGRs) and the principles of corporate governance for banks operating in Saudi Arabia (PCGB) regarding board characteristics and provide insights to improve board effectiveness and corporate governance practice in general.

**Keywords:** corporate governance; board education; board diversity; board IT experience; Saudi Arabia

**JEL Classification:** G3; G21; L25; O16

## 1. Introduction

Corporate governance is the system of rules, practices, and processes used to direct, manage, and control a company. Good governance is important because it provides the infrastructure to improve the quality of the decisions made by corporate management (Chartered Governance Institute UK & Ireland 2022).

A company's board of directors plays a pivotal role in governance as the primary force influencing corporate governance. Therefore, boards of directors are responsible for the governance of their companies.

It is argued that the Asian financial crisis in 1997 was a result of a loss of investor confidence due to the lack of an effective governance system and transparency (Ho and Wong 2001). Moreover, the global financial crisis in 2008 and financial scandals, such as the Enron crisis in 2001 and WorldCom in 2002 followed by the latest scandal related to Wirecard company in 2020, reveal the failure of the board of directors to monitor executive management and protect shareholders' rights.

As a result of these crises and scandals and ongoing concerns about corporate governance quality, boards of directors became the center of the policy debate related to governance reform and the focus of considerable academic research (Adams et al. 2010).

Boards of directors have certain attributes affecting their effectiveness and efficiency, such as board size, independence, CEO duality, and managerial ownership (Vo and Nguyen 2014; Vo and Phan 2013; Isa and Muhammad 2015). The potential role of such common characteristics on firm performance has been mostly investigated in developed countries, and the evidence is inconclusive. Moreover, board characteristics include various attributes, any of which can affect board effectiveness and thus firm performance. These attributes include board education, diversity, experience, attendance, and directorship, which have received less attention by researchers and empirical studies, notably in developing countries.

This study, therefore, aims to investigate the potential effects of a set of board characteristics, including board size, board composition (independent and executive board members), board education (educated board members, board members with Masters' degrees, and with PhD degrees), board diversity (gender and nationality), board meeting attendance, multiple directorships, and board experience (board members with IT experience) on Saudi bank performance in light of Saudi corporate governance regulations.

The focus on the role of board characteristics on Saudi bank performance is motivated by certain issues. First, this study is encouraged by the call of Dalwai et al. (2015) and Almoneef and Samontaray (2019) for further research on corporate governance mechanisms in the banking sector of Gulf Cooperation Council (GCC) countries, in general, and Saudi Arabia, in particular, to improve governance practices in such a vital financial sector, and to extend the literature.

Second, despite the relatively extensive research on the role of certain common board characteristics, such as board size, independence, non-executives, CEO duality, meeting frequency, and managerial ownership on firm performance (e.g., Conheady et al. 2015; Hoang et al. 2017; Rashid 2018; Almoneef and Samontaray 2019), the potential role of other board characteristics, such as board education, diversity, experience, attendance, and multiple directorships, on firm performance has attracted less attention by researchers, especially in emerging markets (Issa et al. 2021), which necessitates further research.

Third, corporate governance in the Saudi banking sector is highly regulated compared to other financial and non-financial sectors due to the unique role played by banks in the overall economy and in its credit provision and liquidity functions (Abraham 2013). In the wake of the Saudi stock market crisis in 2006, and to restore investor confidence, the Capital Market Authority (CMA) issued the Saudi corporate governance regulations (SCGRs) at the end of 2006 (CMA 2006). The SCGRs include basic rules and standards to regulate listed firms on Tadawul and to ensure adherence to the best corporate governance practices that protect the rights of all shareholders. The SCGRs highlight the importance of the board of directors as the main pillar to ensure best practices of corporate governance. Later, the Saudi Arabian Monetary Agency (SAMA) issued the principles of corporate governance for banks operating in Saudi Arabia (PCGB) in 2014 (SAMA 2014), which emphasize, among others, the critical role of the board of directors and its characteristics, such as size, composition, academic qualifications, experience, attendance, and directorship as good governance mechanisms. Furthermore, the PCGB identify the criteria of best practices of some of these characteristics. However, there is a lack of empirical evidence of whether compliance with these principles related to board characteristics by listed banks leads to better governance practices and thus better performance. This study attempts to answer this question by investigating the relationship between bank performance and board characteristics mentioned in the PCGB, in addition to board diversity, in terms of gender and nationality diversity, which is not clearly mentioned by the PCGB.

The current study differs from prior studies in Arab and GCC countries (e.g., Al-rashed 2010; Azzoz and Khamees 2016; Pillai and Al-malkawi 2018) that investigate corporate governance mechanisms, including certain common characteristics of boards of directors, such as board size, independent and non-executive board members, and CEO duality, and

none of them focus on the banking sector. Moreover, as this study focuses on the Saudi banking sector, it is distinguished from the study by (Elbahar 2019), who examines the relationship between certain board characteristics (i.e., board size, non-executive members on the board, female board member, CEO turnover) and bank performance in GCC countries for four years using two measurements of bank performance (ROA, ROE). Furthermore, Issa et al. (2021) focus mainly on the role of board diversity on financial performance of a sample of banks listed in 11 countries in the Middle East and North Africa (MENA) region (including six banks from Saudi Arabia). Recently, El-Chaarani et al. (2022) investigated the effect of internal and external corporate governance mechanisms, including board size, independence, gender, and CEO duality, on the financial performance of banks in the MENA during the COVID-19 pandemic period.

Moreover, this study differs from previous studies in the Saudi context (Y. A. Al-Matari et al. 2012; Fallatah and Dickins 2012; Ghabayen 2012; Habbash and Bajaher 2015; Al-faryan 2017; Buallay et al. 2017; Abdalkrim 2019; Hamdan et al. 2019; Hamdan 2018), as these studies either exclude the banking sector or test only a few common board characteristics with short time periods of investigation. The relevant studies to the current research are those conducted by Al-Sahafi and Rodrigs (2015) and Almoneef and Samontaray (2019), who investigate the role of corporate governance on Saudi bank performance during a short period of 4 years. They address a few board characteristics, including board size, independence, meeting frequency, CEO duality, and foreign board membership. Moreover, Habtoor (2020) examines the moderating role of ownership concentration on the relationship between board composition and Saudi bank performance. Furthermore, Habtoor (2021) investigates the influence of board ownership on Saudi bank performance. Recently, E. M. Al-Matari et al. (2022) examined the impact of board characteristics (size, independence, financial experience, meetings, and attendance) and Fintech (financial technology) on the performance of financial firms. However, the current study focuses on the impact of 11 board characteristics on Saudi bank performance with a longitudinal data set of 10 years of investigation.

In sum, this study contributes to the existing literature and provides important practical implications. Firstly, the current study attempts to rely on multiple theoretical perspectives drawn from several accounting theories for a deeper understanding of how and which board characteristics could affect bank performance. Secondly, since there is a dearth of research on the role of board characteristics, notably board education, diversity, and experience, on firm performance, this study aims to narrow the literature gap by providing empirical evidence on the influence of such characteristics on bank performance from a unique institutional, cultural, and economic environment. To the best of the author's knowledge, this is the first study that investigates comprehensively the impact of various characteristics of boards of directors on Saudi bank performance. Thirdly, the empirical results highlight the conflicting role of board independence on bank performance and suggest, among others, that the appointment of board members should be strictly based on a higher level of education and experience, notably in IT experience, which would ensure better bank performance. Moreover, the positive role of foreign CEOs on bank performance indicates that more cultural diversity among the board and top management would bring benefits to firms, which may outweigh the potential costs of conflict and communication problems caused by cultural differences. These findings are important for Saudi policymakers and regulatory authorities, including the CMA and SAMA, to assess the current practice of governance in the banking industry and to construct an appropriate set of governance mechanisms. Companies and market participants might use these findings to shape their understanding of the role of boards of directors on bank performance.

The remainder of this study is structured as follows. Section 2 provides the literature review and hypotheses development. Section 3 describes the research methodology. Section 4 reports the empirical results and discussion. Conclusions, limitations, and future research are offered in Section 5.

## 2. Literature Review and Hypotheses Development

The corporate governance literature reveals extensive research on the potential effect of different mechanisms of corporate governance on firm performance (e.g., Afrifa and Tauringana 2015; LaRosa and Bernini 2018; Roudaki 2018; Kao et al. 2019; Iwasaki et al. 2022; Liu et al. 2022; Shakri et al. 2022; Jesuka and Peixoto 2022; Drobetz et al. 2021; Al-Jalahma 2022). However, less attention has been paid to the impact of various characteristics of the board on firm performance (e.g., Amin and Nor 2019; Jensen et al. 2020; Sarhan et al. 2019; Livnat et al. 2021; Guney et al. 2020; Amrani et al. 2022; Andoh et al. 2022; Hamid and Purbawangsa 2022). Moreover, most of this research is conducted in developed countries, and less is done in developing countries, such as Arab countries, GCC countries, and Saudi Arabia, specifically (e.g., Al-Sahafi and Rodrigs 2015; Azzoz and Khamees 2016; Pillai and Al-malkawi 2018; Abdalkrim 2019; Hamdan et al. 2019; Y. A. Al-Matari 2022; E. M. Al-Matari et al. 2022). Furthermore, the majority of empirical studies focus on prevalent mechanisms of corporate governance and exclude the banking sector (Almoneef and Samontaray 2019). Consequently, and drawing on insights from a number of accounting theories, including agency, resource dependency, stakeholder, and legitimacy theories, supplemented by the implications of Saudi context, this study investigates the potential influence of various characteristics of board of directors, such as board size, board composition (independent and executive board members), board education (educated board members, board members with a Master's degree, board members with a PhD degree), board diversity (gender and nationality), board experience (board members with IT experience), board meeting attendance, and multiple directorships on Saudi bank performance.

### 2.1. Board Size and Bank Performance

Agency theory suggests that board size is an important element affecting board effectiveness in monitoring executive management (Jensen 1993). However, the role of size on board effectiveness and thus on firm performance can be explained from two different perspectives. Resource dependency theory suggests that larger boards are more likely to have more qualified members with experience, skills, and diverse views, which can enhance firm performance. In contrast, the organizational behavior research suggests that productivity is negatively related to the size of the working groups (Hackman 1990). This implies that larger boards are more likely to become less effective in performing their duties towards better firm performance because of coordination and communication problems. Empirical evidence reveals mixed results. For instance, Hamid and Purbawangsa (2022); Al-Farooque et al. (2020); and Almoneef and Samontaray (2019) find a positive association between board size and firm performance, while the findings of Dodd and Zheng (2022); Switzer and Tang (2009); and Zabri et al. (2016) reveal a negative impact of board size on firm performance. However, Mihail et al. (2021); Habbash and Bajaher (2015); Amin and Nor (2019); and Y. A. Al-Matari (2022) find no relationship between the two variables.

In the Saudi context, the SCGRs require Saudi listed companies to have boards with no less than three members and no more than eleven, while the PCGB indicate that the appropriate number of board members in the banking sector is between nine and eleven. However, the evidence indicates that boards of Saudi companies are oversized (Al-Abbas 2009; Al-Janadi et al. 2013; Habtoor and Ahmad 2017), as the average number of board members exceeds eight directors (Jensen 1993; Lipton and Lorsch 1992). Moreover, the appointment of board members is more likely to be affected by tribal and social factors and may reflect the controlling shareholders' preferences who hire their relatives and friends. This may suggest that such boards are more likely to be affected by courtesy, favoritism, and politeness at the expense of truth and frankness in the boardroom, which may weaken the board effectiveness and make it easier to be controlled by the CEO or other controlling shareholders (Habtoor and Ahmad 2017) and thus affect negatively bank performance. Accordingly, a hypothesis can be formulated as follows.

**Hypothesis 1 (H1).** *There is a negative relationship between board size and bank performance.*

### 2.2. Board Composition and Bank Performance

Based on agency theory perspective, a higher proportion of independent members on a board of directors would enhance board effectiveness in monitoring executives and limit managerial opportunism. Therefore, independent members on the board can be viewed as a key indicator of corporate governance quality as they are, at least in theory, free from business and other relationships with management, which could materially interfere with the exercising of their independent judgment (Abraham and Cox 2007). In line with the agency theory perspective, the integrated view of resource dependence-legitimacy theories about the influence of board independence on firm performance also considers independent directors on the board as a strategic resource to enhance firm performance through linking the company to an external environment, securing critical resources, reducing environmental dependency, and aiding in establishing and supporting legitimacy (Daily and Dalton 1994). However, stewardship theory and institutional theory adopt an opposite view that higher representation of independent members on the board, as outsiders, would negatively affect the firm strategies and performance, as they are (compared to insiders or executive members) unaware of the strengths and weaknesses of the firms, and unqualified to provide useful counsel and make efficient decisions (Davis et al. 1997; Gaur et al. 2015).

Empirically, El-Chaarani et al. (2022); Y. A. Al-Matari (2022); Al-Farooque et al. (2020); and Villanueva-Villar et al. (2016) demonstrate a positive effect of board independence on firm performance. On the other hand, Fariha et al. (2022); Kumar and Singh (2013); Amin and Nor (2019); and Waheed and Malik (2019) report a negative association between board independence and firm performance. Furthermore, Ararat and Yurtoglu (2021); Carter et al. (2010); Arosa et al. (2013); and Roudaki (2018) fail to find a significant association between board independence and firm performance.

In the Saudi context, the SCGRs emphasize the important role of board independence as a vital tool to protect shareholders' interests and enhance transparency and performance. Moreover, the PCGB indicate that at least two board members must be independent. Thus, having a higher proportion of truly independent members on Saudi companies' boards is essential to improve board effectiveness in monitoring executive management and protecting shareholders' rights against managerial opportunism and wealth expropriation by Saudi-controlling shareholders. Hence, it can be hypothesized that:

**Hypothesis 2 (H2).** *There is a positive relationship between board independence and bank performance.*

On the other hand, agency theory suggests that the presence of executive directors on the board would exacerbate agency conflicts as they have an opportunity to act opportunistically to maximize their own benefits at the expense of the firm owners. However, stewardship theory provides a countervailing view that there is no conflict of interest between executive managers and firm owners, and the main objective of corporate governance is to ensure the optimum composition of the board of directors to achieve board effectiveness. Accordingly, executive board members, as essential parts of team players, are not opportunistic agents but good stewards who act in the best interests of shareholders and the firm (Donaldson and Davis 1991).

Empirical evidence on the relationship between executive directors on the board and firm performance remains scarce. For example, Abraham and Cox (2007) report a positive effect of executive members on the board and corporate risk disclosure. However, Habtoor and Ahmad (2017) find an insignificant association between executive members on the board and corporate risk disclosure. On the other hand, Tenuta and Cambrea (2022) document a negative impact of executive directors on performance of family firms.

In Saudi Arabia, the highly concentrated ownership and the dominance of family ownership as controlling shareholders play essential roles in determining board composition (Habtoor 2020), as they can appoint themselves as CEO or as executive board members

to better serve their interests at the expense of the minority shareholders and other stake-holders. Therefore, the PCGB require that the number of executive members in the board shall not exceed two members. Accordingly, and in line with the agency theory perspective, which is more applicable to the Saudi context, a hypothesis can be formulated as follows:

**Hypothesis 3 (H3).** *There is a negative relationship between executive members on the board and bank performance.*

### 2.3. Board Education and Bank Performance

Education is a type of board general human capital (Li and Patel 2019). In business environments, education is an indicator of various cognitive orientations of a person that can affect firm performance. From a resource dependency theory perspective, educated directors on the board are an important resource of a company, as they are more likely to acquire finer training, better technical expertise, and distinctive skills that enable them to easily understand and analyze the company business environment and then propose solutions to complex issues and strengthen future firm performance (Johnson et al. 2013). In contrast, education may adversely affect firm performance when education gives rise to an exaggeration of one's actual ability as a result of overconfidence biases (Kaur and Singh 2019).

There is a dearth of empirical studies on the role of board education on firm perfor-mance (Fernanaez-Temprano and Tejerina-Gaite 2020). Khidmat et al. (2020) find that board education have a significant positive impact on the accounting and marketing measures of firm performance. Kim and Lim (2010) find a positive impact of education on firm performance. Bennouri et al. (2018) find a positive association between female education and ROA; nevertheless, this association becomes negative with Tobin's Q. However, the results of studies by Fernanaez-Temprano and Tejerina-Gaite (2020) and Boadi and Osarfo (2019) suggest a negative effect of education on firm performance. On the other hand, Assenga et al. (2018) and Issa et al. (2021) find no significant influence of board education on firm performance. While the PCGB refer to board education, they do not identify the specific level of education of a board member. Therefore, and in line with the resource dependency theory perspective, this study examines the potential positive influence of different categories of board education on bank performance as follows:

**Hypothesis 4 (H4).** *There is a positive relationship between the number of educated board members and bank performance.*

**Hypothesis 5 (H5).** *There is a positive relationship between board members with a Master's degree and bank performance.*

**Hypothesis 6 (H6).** *There is a positive relationship between board members with a PhD degree and bank performance.*

### 2.4. Board Diversity and Bank Performance

There are many proxies of board diversity, such as gender, nationality, ethnicity, and race. Agency theory links board diversity with lower agency conflicts because diversified boards are likely to be more independence and more effective in monitoring management, which enhance firm performance (Safiullah et al. 2022; J. Singh et al. 2022; Abdullah 2014). Moreover, resource dependency theory considers board diversity as a key human capital resource that can bring different personalities, perspectives, proficiencies, experience, and capabilities to link the firm to external environment and facilitate the access to different national and international markets, which can enhance geographic and product diversifica-tion and thus improve firm performance (Amin and Nor 2019; Sarhan et al. 2019; Safiullah et al. 2022; Dodd and Zheng 2022). As an extension to this view, stakeholder theory and legitimacy theory argue that diversified boards, in terms of race, ethnicity, nationality,



gender, age, and education, are more likely to include different groups of stakeholders and represent different perspectives of essential components of society. Such diversity enhances board effectiveness to monitor and enforce management to act in accordance with all stakeholders' rights and contributes to legitimize the firm's strategies and operations. Moreover, the social psychological theory argues that the effect of board diversity on firm performance could be positive or negative. Diversified boards introduce diversified ideas, divergent perspectives, critical thinking, and innovations that can be sifted and adapted to enhance firm performance (Westphal and Milton 2000). However, excessive diversity in terms of gender, race, and nationality may create heterogeneous working groups with different backgrounds and perceptions, which may lead to communication problems and delays in decision-making processes, thus impairing firm performance (Delis et al. 2017; Amin and Nor 2019; Salloum et al. 2019) Furthermore, board diversity may be positively correlated with multiple directorships, which would lead to board busyness. In such cases, the costs of busyness may outweigh the benefits of diversity.

Prior empirical studies report inconclusive evidence on the influence of different proxies of board diversity and firm performance. In fact, the progressive inclusion of women into companies' boards and managerial positions has been one of the relevant aspects of good corporate governance practice (Boadi et al. 2022). Fariha et al. (2022) find that firms with higher board gender diversity achieve better accounting-based firm performance but lower market-based firm performance. Hassan and Marimuthu (2018) report a positive association between gender diversity and firm performance. However, they find no impact of ethnic diversity on firm performance. Elsharkawy et al. (2018) find that non-national directors are negatively related to firm performance. Sarhan et al. (2019) indicate that board diversity, as measured by gender and nationality, is related to higher firm financial performance. However, M. Adams and Baker (2021) document an insignificant effect of foreign CEOs on ROE. Furthermore, Amin and Nor (2019) report no impact of board ethnic diversity on firm performance. Moreover, Amrani et al. (2022); El-Chaarani et al. (2022); and Fernanaez-Temprano and Tejerina-Gaite (2020) find an insignificant influence of board gender on firm performance.

In Saudi-listed companies in general and in the banking sector in particular, female representation and thus participation in board decisions is very limited. However, recently, Saudi women have been freed from many restrictions, especially in light of the vision of 2030, which gives women the right to participate in various aspects of life. Despite the Saudi government and related agencies encouraging and supporting Saudi women to exercise their rights in leadership and management, however, the governance regulations are devoid of any reference to the role of women as a tool for strengthening governance. Based on theoretical perspectives and empirical evidence, a hypothesis can be formulated as follows.

**Hypothesis 7 (H7).** *There is a significant relationship between gender diversity and bank performance.*

Furthermore, ownership in Saudi-listed companies is highly concentrated with a dominance of Saudi-controlling families, which has a negative influence on corporate board decisions related to transparency, disclosure, and performance (e.g., Habtoor et al. 2019; Habtoor 2020). However, this study assumes that nationality diversity proxied by CEO nationality would enhance bank performance, because foreign chief executive officers are more likely to be free from tribal and social relations and may acquire finer training and richer experience to propose solutions to complex issues and have connections that facilitate the access to various resources, which would enhance bank performance. Accordingly, it can be hypothesized that:

**Hypothesis 8 (H8).** *There is a positive relationship between nationality diversity of the CEO and bank performance.*

*2.5. Board Meeting Attendance and Bank Performance*

Board meetings are the most usual occasions for board members to discuss and exchange ideas about monitoring managers and executing other board duties (De Andres et al. 2005). Therefore, the attendance level of board meetings is an important indicator of board diligence and efficiency. Brown and Caylor (2006) indicate that board meeting attendance is one of the seven most significant governance measures affecting firm performance in the U.S.

Theoretically, agency theory suggests that board members who attend more board meetings are more likely to better perform their duties in monitoring executive management and protecting shareholders' rights. Moreover, resource dependency theory argues that the commitment and involvement of board members to attend board meetings would provide a better working environment with different and updated experiences, views, and opinions shared among executive and non-executive directors, which enhance board diligence and effectiveness to act in accordance with the interests of shareholders.

Empirical studies mostly focus on board meeting frequency as a proxy for board diligence and activity (e.g., Villanueva-Villar et al. 2016; Habtoor and Ahmad 2017; Al-Farooque et al. 2020; Fariha et al. 2022; Y. A. Al-Matari 2022). However, less interest is given to board meeting attendance. For example, E. M. Al-Matari et al. (2022) find that Fintech has significant influence on firm performance. Bhatt and Bhattacharya (2015) indicate that firm performance is positively affected by board meeting attendance. Moreover, Chou et al. (2013) find that board meeting attendance by directors themselves is positively related to firm performance, while attendance by directors' representatives has a negative impact on firm performance. Furthermore, Gray and Nowland (2018) demonstrate that when firms hold additional board meetings, lower director attendance is related to lower firm performance. Therefore, the SCGRs emphasize the importance of meeting attendance by board members. More specifically, the PCGB states that "all members should attend and participate in board meetings of the bank, and if a member fails to attend three meetings a year without an excuse, he/she should be substituted by another member". Based on the theoretical arguments, empirical evidence, and Saudi governance regulations, this study proposes that:

**Hypothesis 9 (H9).** *There is a positive relationship between board meeting attendance and bank performance.*

*2.6. Board Directorship and Bank Performance*

Multiple directorships are an important characteristic affecting board effectiveness and thus firm performance. However, the existing literature on the role of interlocking directorships is inconclusive, and empirical evidence is conflicted. Theoretically, resource dependency theory and the reputation hypothesis consider multiple directorships as a vital source to improve board reputation, which in turn enhance firm performance. Holding multiple directorships by board members would allow them to acquire new and different managerial, economic, and social experiences and skills (Roudaki and Bhuiyan 2015; Ferris et al. 2020). On the other hand, the busyness hypothesis suggests that holding multiple directorships increases the workload of board of directors, which may impair board effectiveness and thus firm performance. In sum, at a lower level of directorships, the benefits of accumulated resources of holding an acceptable number of interlocking directorships outweigh the costs of board busyness. However, the costs of holding too many memberships would outweigh the benefits of accumulated resources as the board becomes too busy and overworked, which affect negatively the monitoring quality and thus firm performance.

In the same vein, empirical evidence reveals mixed results. Some studies document a positive association between multiple directorships and firm performance (e.g., Ferris et al. 2020; Kaur and Singh 2019; Song et al. 2021). However, other studies report a negative impact of multiple directorships on firm performance (e.g., Roudaki and Bhuiyan 2015;



Latif et al. 2020; Nam and An 2018), while Ferris et al. (2003) and (Devos et al. 2009) find no evidence of a relationship between the two variables.

In the Saudi context, the SCGRs and the PCGB realize the potential negative impact of excessive multiple directorships by board members and therefore restrict the number of memberships by board member to no more than five listed companies at the same time. Accordingly, it can be hypothesized that:

**Hypothesis 10 (H10).** *There is a significant relationship between multiple directorships and bank performance.*

*2.7. Board IT Experience and Bank Performance*

Information technology (IT) has become an important indicator for success in modern organizations and a critical factor of their survival and growth (E. M. Al-Matari et al. 2022). IT governance controls refer to the administrative, operational, and technical procedures or countermeasures prescribed in the information system to protect the confidentiality, integrity, and availability of the system and its information (Hamdan et al. 2019). Therefore, IT governance is an integral part of the functions and responsibilities of companies' boards and executive managements to enable them to manage risks and evaluate the efficiency of IT investment, which in turn reflects positively on firm performance (Benaroch and Fink 2021).

Resource dependency theory has a proper explanation and justification for the potential positive influence of board experience on firm performance. This theory considers that experienced members on the board are vital human capital assets that provide the firm with a vast amount of accumulated experience, skills, and knowledge that can be utilized in monitoring management and leading the firm towards better performance. Furthermore, IT experience and awareness is an important element of board experience to enhance firm value and performance. Accordingly, board members with higher levels of IT experience and awareness are more likely to understand, manage, and utilize IT governance and related resources to enhance board oversight and create competitive advantages, which would lead to better performance (Zhang et al. 2018).

Despite the scarcity of research on this issue, empirical evidence largely supports the significant influence of IT experience in enhancing firm performance (e.g., Zhang et al. 2018; Erkmen et al. 2020; Hamdan et al. 2019; E. M. Al-Matari et al. 2022).

Recently, Saudi Arabia witnessed an economic opening in line with the kingdom's Vision 2030. This would attract a large number of international and multination companies, together with the technology and advanced technical and managerial experience they have, which may put Saudi companies in fierce competition with foreign companies if they do not absorb or keep pace with the development in information technology. In this regard, the PCGB recognize the importance of board experience as a critical source of board competency and require board members of the bank to have "diversified experience of no less than ten years in different areas institutions position jurisdictions, such as banking, insurance, business, economics, and accounting". However, it is important for Saudi regulatory bodies to regulate the IT governance in Saudi companies and ensure that corporate boards include experienced members in IT. Accordingly, this study attempts to support the argument above by empirically examining the potential effect of board IT experience on bank performance in an industry that depends heavily on advanced technology, including IT. Accordingly, a hypothesis can be formulated as follows:

**Hypothesis 11 (H11).** *There is a positive relationship between IT experience of board members and bank performance.*

## 3. Research Methodology

### 3.1. Sample Selection and Data Collection

To test the hypotheses, the sample of this study was drawn from the annual reports of Saudi banks listed on Tadawul over the period of 2009–2018. Non-financial firms were excluded since they are less-regulated, and applied accounting standards are different from those of banks. Moreover, other financial firms and non-listed banks were also dropped from the sample because they are less committed to corporate governance regulations, in addition to the lack of data on variables of the study. Furthermore, out of the 120 firm-year observations, 30 observations were also excluded from the sample due to missing data for certain independent and control variables. This exclusion resulted in a final sample of 90 bank-year observations. Data were collected from banks' annual reports downloaded from Tadawul and bank websites.

### 3.2. Definition of Variables and Model Specification

The current study classified variables involved in the regression models into three main categories, namely, dependent variables, independent variables, and control variables, with full definitions as shown in Table 1.

**Table 1.** Definition and measurement of variables.

| Notation | Variable Name | Description/Measurement |
|---|---|---|
| Dependent variable (bank performance) | | |
| ROA | Return on assets | Net income divided by book value of total assets |
| ROE | Return on equity | Net income divided by book value of total equities |
| Q | Tobin's Q | Market value of total shares plus book value of debt divided by book value of assets |
| Independent variables (board characteristics) | | |
| B_SIZE | Board size | Number of board members |
| B_IND | Board independence | Percentage of independent members on the board |
| B_EXE | Executive members on the board | Percentage of executive members on the board |
| B_EDU | Educated board members | Number of board members with at least a Bachelor's degree |
| B_MASTER | Number of board members with a Master's degree | Number of educated board members with a Master's degree as maximum |
| B_PHD | Number of board members with a PhD degree | Number of educated board members with a PhD degree |
| GENDER | Gender diversity | Dummy variable of 1 if the board contains a female member, and 0 otherwise |
| CEO_NAT | CEO nationality diversity | Dummy variable of 1 if the CEO is foreign, and 0 otherwise |
| B_ATTEND | Board meeting attendance | Average of board meetings attendance by board members per year |
| B_DIRECT | Multiple directorships of board members | Number of multiple directorships of board members per year |
| B_IT_EXP | Board members with IT experience | Percentage of board members with information technology experience and knowledge |
| Control variables (firm characteristics and ownership structure) | | |
| F_SIZE | Bank size | Total bank employees |
| F_AGE | Bank age | Number of years the bank has been established |
| F_LEV | Leverage | Ratio of total debt to total assets |
| O_CONCEN | Ownership concentration | The percentage of bank shares held by large shareholders who hold 5% and above of bank shares |

First, the dependent variable was bank performance, which was measured by the most common three different measures: Return on Assets (ROA) as an operational performance measure, Return on Equity (ROE) as a financial performance measure, and Tobin's Q (Tobin's Q) as a market-based performance measure (see Table 1).

Second, to test the main hypotheses (H1–H11) related to board characteristics, the independent variables included board size, board composition (board independence, executive members on the board), board education (educated board members, board members with a Master's degree, board members with a PhD degree), board diversity (gender diversity, CEO nationality diversity), board meeting attendance, board IT experience, and multiple directorships (see Table 1).

Third, to control for potential omitted variable bias (Gujarati 2003; Wooldridge 2002), and to rule out alternative explanations for the mean results (J. V. Singh et al. 1986), this study used a number of control variables, including firm-specific characteristics (i.e., firm size, firm age, leverage) and ownership structure, such as ownership concentration (see Table 1). For brevity, this study did not develop direct theoretical relationships between control variables and bank performance, because there are extensive theoretical and empirical literatures that suggest a significant influence on firm performance (e.g., Giraldez-Puig and Berenguer 2018; Bennouri et al. 2018; Zhang et al. 2018; Boadi and Osarfo 2019; Hamdan et al. 2019; Waheed and Malik 2019; Fernanaez-Temprano and Tejerina-Gaite 2020; Latif et al. 2020; Farooq et al. 2022; Y. A. Al-Matari 2022; Chatterjee and Nag 2022; Hamid and Purbawangsa 2022; Jesuka and Peixoto 2022; Uyar et al. 2022; Fariha et al. 2022).

### 3.3. Data Analysis

Given the panel nature of the data, this study attempted to employ unbalanced panel data analysis using STATA software. To identify whether the ordinary least squares (OLS) or panel data (fixed and/or random effects) technique were more appropriate to analyze the data set, the Lagrange Multiplier (LM) test (Breusch and Pagan 1980) was applied to test the presence of random effects by comparing the random effects model with the OLS model. As shown in Table 2, the results of the LM test were insignificant for all models of study (see Table 2), and thus the null hypothesis of no random effects could not be rejected, which means that the application of the OLS was more appropriate than random effects techniques to analyze the data set. Furthermore, Hutcheson and Sofroniou (1999) argue that when the regression model includes continuous and dummy or binary variables, the OLS estimation is more appropriate than fixed effects.

**Table 2.** Diagnostic Tests.

| Diagnostic Tests | Model (1) ROA | Model (2) ROE | Model (3) Q |
|---|---|---|---|
| Lagrange Multiplier (LM) test for random effects (*p*-value) | 1.000 | 1.000 | 1.000 |
| Shapiro–Wilk test for normality (*p*-value) | 0.581 | 0.111 | 0.661 |
| Ramsey test (*p*-value) | 0.600 | 0.864 | 0.867 |
| Modified Wald test for heteroskedasticity (*p*-value) | 0.000 | 0.000 | 0.039 |
| Wooldridge test for autocorrelation (*p*-value) | 0.000 | 0.007 | 0.000 |
| Durbin–Wu–Hausman test for endogeneity (*p*-value) | 0.174 | 0.208 | 0.001 |

Prior to analysis, the main assumptions of multiple regression analysis, such as outliers, normality, linearity, multicollinearity, heteroskedasticity, and autocorrelation, were checked and then corrected or controlled. Normality tests of dependent and continuous independent and control variables using Shapiro–Wilk test indicated that the data were not normally distributed (for brevity not reported here, but available upon request). Accordingly, the dependent and continuous variables were transformed into normal scores using the Van der Waerden approach as it transforms actual observations to their equivalent values on the normal distribution and minimizes the effect of outliers (Cooke 1998). After transforming data, the normality assumption was re-examined, and the results of Shapiro–Wilk test

became insignificant for all models (see Table 2), which indicated that the null hypothesis that the data is normally distributed could not be rejected. This indicated that the use of normal scores, instead of original ones, in the models would produce more robust results.

To check for non-linearity, the results of the scatter plots indicated no clear departure from linearity (for brevity not reported here, but available upon request). Moreover, the Ramsey test was conducted, and the results were insignificant (see Table 2), indicating appropriate specification of all models.

Multicollinearity was also checked using a Pearson correlation matrix (see Table 3) and variance inflation factor (VIF) (see Table 4). The results indicated no severe multicollinearity problem, as shown in Tables 2 and 3, respectively, since maximum values did not exceed the threshold value of correlation (0.80) and the VIF (10) (Gujarati 2003; Hair et al. 2010).

However, additional analysis was done (for brevity not reported here, but available upon request) by excluding the highest two correlated variables (B_SIZE and B_EDU) from regression models to determine whether their presence had a significant impact on the results of other variables involved in the models. The exclusion of the two variables from the models alternately in the first two steps, and the exclusion of both of them in the third step had almost no effect on the regression results of the remaining variables. This means that B_SIZE and B_EDU variables had no major influence on the results of other variables in the models. Moreover, multicollinearity was re-examined after each step of exclusion, and the results became much better, as the highest correlation was significantly reduced to be around 50, and the highest VIF value was below 3, as shown in columns 2–4 in Table 4.

Furthermore, heteroskedasticity is another important assumption to be checked using the Modified Wald test for group-wise heteroskedasticity. The results were significant for all models (see Table 2), which indicated the presence of a heteroskedasticity problem. Autocorrelation was the last assumption that was tested using Wooldridge test (Wooldridge 2002), and the results were significant (see Table 2), indicating the presence of serial correlations for all models of the study, which needed to be corrected or controlled.

Wooldridge (2002) considers that the feasible generalized least squares (FGLS) method is useful to control heteroskedasticity and autocorrelation. Accordingly, and following previous studies (e.g., Emudainohwo 2021; Hoang et al. 2017; Nguyen 2020; Vo and Phan 2013), this study employed the FGLS estimation as the main method to analyze the relationship between board characteristics and bank performance as follows:

$$
\begin{aligned}
\text{PERF} = {} & \beta_0 + \beta_1 \text{B\_SIZE}_{it} + \beta_2 \text{B\_IND}_{it} + \beta_3 \text{B\_EXE}_{it} + \beta_4 \text{B\_EDU}_{it} + \beta_5 \text{B\_MASTER}_{it} \\
& + \beta_6 \text{B\_PHD}_{it} + \beta_7 \text{GENDER}_{it} + \beta_8 \text{CEO\_NAT}_{it} + \beta_9 \text{B\_ATTEND}_{it} + \beta_{10} \text{B\_DIRECT}_{it} \\
& + \beta_{11} \text{B\_IT\_EXP}_{it} + \beta_{12} \text{F\_SIZE}_{it} + \beta_{13} \text{F\_AGE}_{it} + \beta_{14} \text{F\_LEV}_{it} + \beta_{15} \text{O\_CONCEN}_{it} + \\
& \varepsilon_{it}
\end{aligned}
\tag{1}
$$

where PERF is the bank performance (dependent variable), which is measured using both accounting-based measures (i.e., return on assets ROA and return on equity ROE) and market-based measure (Tobin's Q); B_SIZE is board size; B_IND is board independence; B_EXE is executive members on the board; B_EDU is educated board members; B_MASTER is board members with a Master's degree; B_PHD is board members with a PhD degree; GENDER is gender diversity; CEO_NAT is CEO nationality diversity; B_ATTEND is board meeting attendance; B_DIRECT is multiple directorships of board members; B_IT_EXP is board members with IT experience; F_SIZE is bank size; F_AGE is bank age; F_LEV is bank leverage; O_CONCEN is ownership concentration; and $\varepsilon$ is an error term.

Despite this study relying on the FGLS as the main method of analysis, the OLS and random effects regression models were used for comparison and robustness check purposes, while the panel-corrected standard errors (PCSE) and the two-stage least squares (2SLS) models were used for additional analysis.

**Table 3.** Pearson correlation matrix.

| VARIABLES | 1 | 2 | 3 | 4 | 5 | 6 | 7 | 8 | 9 | 10 | 11 | 12 | 13 | 14 | 15 | 16 | 17 | 18 |
|---|---|---|---|---|---|---|---|---|---|---|---|---|---|---|---|---|---|---|
| 1. ROA | 1 | | | | | | | | | | | | | | | | | |
| 2. ROE | 0.756 *** | 1 | | | | | | | | | | | | | | | | |
| 3. Tobin's Q | 0.423 *** | 0.416 ** | 1 | | | | | | | | | | | | | | | |
| 4. B_SIZE | 0.267 *** | 0.303 ** | 0.381 *** | 1 | | | | | | | | | | | | | | |
| 5. B_IND | −0.070 | −0.227 ** | 0.074 | 0.021 | 1 | | | | | | | | | | | | | |
| 6. B_EXE | 0.141 | 0.180 * | 0.053 | −0.248 ** | −0.219 ** | 1 | | | | | | | | | | | | |
| 7. B_EDU | 0.081 | 0.084 | 0.170 * | 0.750 *** | −0.125 | −0.262 * | 1 | | | | | | | | | | | |
| 8. B_MASTER | 0.483 *** | 0.379 ** | 0.292 *** | 0.376 *** | −0.134 | 0.138 | 0.317 *** | 1 | | | | | | | | | | |
| 9. B_PHD | −0.154 | −0.312 ** | −0.100 | −0.134 | 0.285 *** | −0.015 | 0.115 | −0.172 * | 1 | | | | | | | | | |
| 10. GENDER | −0.073 | 0.094 | −0.064 | 0.053 | −0.120 | 0.051 | 0.012 | 0.219 ** | 0.064 | 1 | | | | | | | | |
| 11. CEO_NAT | 0.337 *** | 0.460 ** | 0.106 | 0.305 *** | −0.050 | 0.329 * | 0.043 | 0.393 *** | −0.254 *** | 0.333 *** | 1 | | | | | | | |
| 12. B_ATTEND | 0.198 ** | 0.233 ** | −0.030 | 0.117 | −0.147 | −0.103 | 0.350 *** | 0.211 ** | 0.032 | −0.046 | −0.051 | 1 | | | | | | |
| 13. B_DIRECT | 0.036 | 0.173 * | 0.055 | 0.273 *** | −0.376 *** | −0.223 * | 0.373 *** | 0.103 | −0.425 *** | −0.092 | −0.006 | 0.184 * | 1 | | | | | |
| 14. B_IT_EXP | 0.179 * | 0.054 | −0.044 | −0.117 | −0.039 | −0.001 | −0.279 *** | −0.031 | −0.438 *** | −0.151 | −0.224 ** | −0.133 | −0.083 | 1 | | | | |
| 15. F_SIZE | 0.492 *** | 0.408 ** | 0.323 *** | 0.425 *** | −0.133 | −0.083 | 0.167 | 0.279 *** | −0.329 *** | −0.295 *** | 0.120 | 0.203 * | 0.389 *** | 0.166 | 1 | | | |
| 16. F_AGE | −0.175 * | −0.030 | −0.415 *** | −0.179 * | −0.035 | −0.218 * | −0.012 | −0.134 | −0.147 | 0.188 * | 0.040 | 0.270 *** | 0.078 | 0.076 | −0.121 | 1 | | |
| 17. F_LEV | −0.140 | 0.448 ** | 0.165 * | 0.068 | −0.270 *** | 0.057 | −0.025 | −0.143 | −0.293 *** | 0.240 ** | 0.174 * | 0.058 | 0.182 * | −0.064 | −0.092 | 0.282 *** | 1 | |
| 18. O_CONCEN | 0.224 ** | 0.283 ** | 0.002 | 0.229 ** | −0.043 | −0.138 | 0.268 *** | 0.518 *** | 0.017 | 0.380 *** | 0.336 *** | 0.246 ** | 0.094 | −0.149 | 0.167 | 0.349 *** | 0.119 | 1 |

ROA is return on assets; ROE is return on equity; Q is Tobin's Q; B_SIZE is board size; B_IND is board independence; B_EXE is executive members on the board; B_EDU is educated board members; B_MASTER is board members with a Master's degree; B_PHD is board members with a PhD degree; GENDER is gender diversity; CEO_NAT is CEO nationality diversity; B_ATTEND is board meeting attendance; B_DIRECT is multiple directorships of board members; B_IT_EXP is board members with IT experience; F_SIZE is bank size; F_AGE is bank age; F_LEV is bank leverage; O_CONCEN is ownership concentration. Notes: * Significant at the 10% level. ** Significant at the 5% level. *** Significant at the 1% level.

**Table 4.** Variance inflation factor (VIF).

| All Variables (1) | | All Variables Are Included Except B SIZE (2) | | All Variables Are Included Except B EDU (3) | | All Variables Are Included Except B SIZE and B EDU (4) | |
|---|---|---|---|---|---|---|---|
| **Variable** | **VIF** | **Variable** | **VIF** | **Variable** | **VIF** | **Variable** | **VIF** |
| B_SIZE | 8.14 | | | | | | |
| B_EDU | 7.03 | B_MASTER | 2.905 | B_MASTER | 2.91 | | |
| B_MASTER | 2.92 | B_PHD | 2.702 | CEO NAT | 2.64 | B_MASTER | 2.78 |
| CEO NAT | 2.88 | O_CONCEN | 2.436 | O_CONCEN | 2.48 | O_CONCEN | 2.43 |
| B_PHD | 2.84 | CEO NAT | 2.409 | B_PHD | 2.48 | B PHD | 2.39 |
| F_SIZE | 2.75 | B_IT EXP | 2.176 | B_EXE | 2.20 | CEO NAT | 2.37 |
| O_CONCEN | 2.55 | B_DIRECT | 2.104 | B_IT EXP | 2.17 | B_IT EXP | 2.17 |
| B_EXE | 2.23 | B_EXE | 2.026 | F_SIZE | 2.16 | F_SIZE | 1.98 |
| B_DIRECT | 2.21 | F_SIZE | 1.984 | B_SIZE | 2.01 | B_DIRECT | 1.93 |
| B_IT EXP | 2.20 | GENDER | 1.869 | B_DIRECT | 1.97 | GENDER | 1.86 |
| GENDER | 2.03 | B_EDU | 1.734 | GENDER | 1.88 | B_EXE | 1.84 |
| F_AGE | 1.89 | B_IND | 1.725 | F_AGE | 1.73 | F_LEV | 1.67 |
| B_IND | 1.89 | F_LEV | 1.681 | F_LEV | 1.68 | B_IND | 1.66 |
| F_LEV | 1.78 | F_AGE | 1.588 | B_IND | 1.66 | F_AGE | 1.57 |
| B_ATTEND | 1.53 | B_ATTEND | 1.516 | B_ATTEND | 1.49 | B_ATTEND | 1.48 |
| Mean VIF | 2.99 | Mean VIF | 2.061 | Mean VIF | 2.10 | Mean VIF | 2.01 |

## 4. Empirical Results and Discussion

### 4.1. Descriptive Analysis

As shown in Table 5, the mean values for the ROA, ROE, and Tobin's Q were 0.018, 0.126, and 1.061, respectively, and ROA ranged from 0.00 to 0.032, and ROE ranged from 0.008 to 0.215, which means that the indicators of bank performance displayed large variations.

**Table 5.** Descriptive statistics.

| Variables | N | MIN | MAX | MEAN | STD. DEV. | Skewness | Kurtosis |
|---|---|---|---|---|---|---|---|
| | | | Dependent variable: Bank performance | | | | |
| ROA | 90 | 0.001 | 0.032 | 0.018 | 0.005 | −0.49 | 4.426 |
| ROE | 90 | 0.008 | 0.215 | 0.126 | 0.034 | −0.319 | 4.491 |
| Tobin's Q | 90 | 0.963 | 1.264 | 1.061 | 0.061 | 1.066 | 4.469 |
| | | | Independent variables: Board characteristics | | | | |
| B_SIZE | 90 | 7 | 11 | 9.767 | 0.78 | −0.572 | 3.895 |
| B_IND | 90 | 0.222 | 1 | 0.479 | 0.154 | 1.331 | 5.261 |
| B_EXE | 90 | 0 | 0.2 | 0.069 | 0.063 | 0.268 | 2.141 |
| B_EDU | 90 | 6 | 11 | 9.122 | 0.992 | −0.593 | 4.196 |
| B_MASTER | 90 | 0 | 8 | 3.844 | 1.748 | −0.154 | 2.671 |
| B_PHD | 90 | 0 | 3 | 1.444 | 1.061 | 0.004 | 1.782 |
| GENDER | 90 | 0 | 1 | 0.089 | 0.286 | 2.889 | 9.348 |
| CEO_NAT | 90 | 0 | 1 | 0.456 | 0.501 | 0.178 | 1.032 |
| B_ATTEND | 90 | 0.81 | 1 | 0.937 | 0.042 | −0.197 | 2.729 |
| B_DIRECT | 90 | 3 | 72 | 23.111 | 14.26 | 1.193 | 4.215 |
| B_IT_EXP | 90 | 0 | 2 | 0.289 | 0.525 | 1.613 | 4.697 |
| | | Control variables: Firm specific characteristics and ownership structure | | | | | |
| F_SIZE | 90 | 1071 | 13,684 | 4304.222 | 3126.749 | 2.032 | 6.182 |
| F_AGE | 90 | 14 | 63 | 39.578 | 12.633 | −0.41 | 3.201 |
| F_LEV | 90 | 0.804 | 0.908 | 0.856 | 0.023 | −0.091 | 2.454 |
| O_CONCEN | 90 | 0.066 | 0.798 | 0.539 | 0.187 | −0.854 | 2.908 |

Regarding board size, the mean was 9.767 members (with a minimum of 7 to a maximum of 11), which was consistent with the average size of the board (10 members) suggested by the PCGB. With respect to board composition, the mean values for board

independence and executive members on the board were 0.479 and 0.069, respectively, and board independence ranged from 0.222 (i.e., two members) to 1.00, and executive members ranged from 0.00 to 0.20 (i.e., two members), which reflected high compliance with the PCGB that state that "The number of executive members in the board shall not exceed two and at least two board members must be independent" (SAMA 2014). For board education, the averages for educated members on the board, board members with a Master's degree, and board members with a PhD were 9.122 (with a minimum of 6 to a maximum of 11), 3.844 (with a minimum of 0 to a maximum of 8), and 1.444 (with a minimum of 0 to a maximum of 3) directors, respectively. The SCGRs and the PCGB stress the importance of academic qualifications without specifying the required educational level. However, the classification of academic qualifications of the board highlights potential conflict of the role of board education on firm performance, which needs further attention from regulatory authorities to strengthen corporate governance. Regarding board diversity, the mean values for the CEO nationality and gender diversity on the board were 0.089 (with a minimum of 0 to a maximum of 1), and 0.456 (with a minimum of 0 to a maximum of 1), respectively. While about half of the CEOs of Saudi listed banks are foreigners, female representation on boards is very low, which may need more attention from regulatory bodies. For board attendance, the mean was 0.937 (with a minimum of 0.81 to a maximum of 1.00), which reflects a high level of meeting attendance by board members as required by the PCGB. The average number of multiple directorships was 23.111 (with a minimum of 3 to a maximum of 72) memberships by board members per year. This average may generally reflect compliance with the SCGRs and PCGB requirements that allow a board member to participate in the membership of other companies' boards, with a maximum of five listed companies. However, a quick scan of the annual reports indicates that there is a considerable number of directors that are also board members in a large number of non-listed companies in addition to their memberships in a maximum five listed companies, which can make them too busy and overworked. Therefore, such an issue should be of interest to regulatory authorities. For IT experience, the mean of board members with IT experience was 0.289 (with a minimum of 0 to a maximum of 2) members, which represents a very low level of IT experience for boards in an industry that depends heavily on advanced technology, including IT.

Regarding control variables, the mean values for bank size, bank age, leverage, and ownership concentration were 4304.222 (with a minimum of 1071 to a maximum of 13,684), 39.578 (with a minimum of 14 to a maximum of 63), 0.856 (with a minimum of 0.804 to a maximum of 0.908), and 0.539 (with a minimum of 0.066 to a maximum of 0.798), respectively.

These findings are of interest to regulatory authorities to determine the extent to which listed banks comply with the SCGRs and the PCGB regarding board characteristics, and provide insights to improve board effectiveness, in particular, and corporate governance, in general.

### 4.2. Univariate Analysis

The correlation matrix in Table 3 presents the potential correlation among variables. This analysis is important as a way to check for multicollinearity and to ensure that the results of multivariate analysis are unbiased (Field 2013).

### 4.3. Multivariate Analysis

As shown in Panel A of Table 6, this study applied the FGLS regression model as the main method of analysis, while the OLS (Panel B) and random effects (Panel C) models were used for comparison and robustness checks purposes.

**Table 6.** Results of FGLS regression and robustness check.

| Variables | Panel A: Main Estimation Using FGLS | | | Panel B: Comparison and Robustness Check OLS | | | RE | | |
|---|---|---|---|---|---|---|---|---|---|
| | Model (1) ROA | Model (2) ROE | Model (3) Q | Model (4) ROA | Model (5) ROE | Model (6) Q | Model (7) ROA | Model (8) ROE | Model (9) Q |
| B_SIZE | 0.312 * | 0.273 | 0.000015 | 0.258 | 0.185 | 0.0502 | 0.258 | 0.185 | 0.0502 |
| | (0.177) | (0.177) | (0.216) | (0.203) | (0.204) | (0.247) | (0.203) | (0.204) | (0.247) |
| B_IND | −0.110 | −0.102 | 0.0777 | −0.137 | −0.111 | 0.150 | −0.137 | −0.111 | 0.150 |
| | (0.0722) | (0.0704) | (0.0944) | (0.0854) | (0.0861) | (0.104) | (0.0854) | (0.0861) | (0.104) |
| B_EXE | −0.0153 | −0.0488 | −0.112 | −0.0196 | −0.0206 | −0.0711 | −0.0196 | −0.0206 | −0.0711 |
| | (0.0818) | (0.0806) | (0.111) | (0.108) | (0.109) | (0.132) | (0.108) | (0.109) | (0.132) |
| B_EDU | −0.499 *** | −0.459 ** | 0.0428 | −0.415 ** | −0.344 * | 0.0484 | −0.415 ** | −0.344 * | 0.0484 |
| | (0.184) | (0.180) | (0.220) | (0.195) | (0.197) | (0.237) | (0.195) | (0.197) | (0.237) |
| B_MASTER | 0.351 *** | 0.278 *** | 0.320 ** | 0.295 *** | 0.247 ** | 0.393 *** | 0.295 *** | 0.247 ** | 0.393 *** |
| | (0.0856) | (0.0803) | (0.130) | (0.108) | (0.109) | (0.132) | (0.108) | (0.109) | (0.132) |
| B_PHD | 0.282 *** | 0.229 ** | 0.0647 | 0.320 ** | 0.296 ** | 0.0597 | 0.320 ** | 0.296 ** | 0.0597 |
| | (0.102) | (0.0957) | (0.148) | (0.130) | (0.131) | (0.158) | (0.130) | (0.131) | (0.158) |
| GENDER | −0.226 | −0.260 | 0.0800 | −0.277 | −0.243 | −0.0275 | −0.277 | −0.243 | −0.0275 |
| | (0.322) | (0.337) | (0.396) | (0.292) | (0.295) | (0.355) | (0.292) | (0.295) | (0.355) |
| CEO_NAT | 0.225 * | 0.290 ** | 0.0177 | 0.307 ** | 0.355 ** | −0.0772 | 0.307 ** | 0.355 ** | −0.0772 |
| | (0.124) | (0.120) | (0.159) | (0.147) | (0.148) | (0.179) | (0.147) | (0.148) | (0.179) |
| B_ATTEND | 0.0592 | 0.145 ** | −0.0167 | 0.0762 | 0.144 * | −0.0321 | 0.0762 | 0.144 * | −0.0321 |
| | (0.0639) | (0.0630) | (0.0800) | (0.0843) | (0.0850) | (0.103) | (0.0843) | (0.0850) | (0.103) |
| B_DIRECT | 0.0665 | 0.104 | −0.00408 | 0.0468 | 0.0678 | −0.0115 | 0.0468 | 0.0678 | −0.0115 |
| | (0.0732) | (0.0709) | (0.101) | (0.0947) | (0.0955) | (0.115) | (0.0947) | (0.0955) | (0.115) |
| B_IT_EXP | 0.282 ** | 0.226 ** | −0.00374 | 0.326 * | 0.234 | −0.00203 | 0.326 ** | 0.234 | −0.00203 |
| | (0.114) | (0.111) | (0.164) | (0.166) | (0.167) | (0.202) | (0.166) | (0.167) | (0.202) |
| F_SIZE | 0.207 ** | 0.120 | 0.294 ** | 0.226 ** | 0.176 * | 0.225 * | 0.226 ** | 0.176 * | 0.225 * |
| | (0.0845) | (0.0810) | (0.115) | (0.103) | (0.104) | (0.126) | (0.103) | (0.104) | (0.126) |
| F_AGE | −0.140 | −0.168 * | −0.395 *** | −0.163 * | −0.213 ** | −0.378 *** | −0.163 * | −0.213 ** | −0.378 *** |
| | (0.0934) | (0.0921) | (0.116) | (0.0929) | (0.0937) | (0.113) | (0.0929) | (0.0937) | (0.113) |
| F_LEV | −0.260 *** | 0.318 *** | 0.329 *** | −0.244 *** | 0.296 *** | 0.385 *** | −0.244 *** | 0.296 *** | 0.385 *** |
| | (0.0775) | (0.0763) | (0.101) | (0.0890) | (0.0897) | (0.108) | (0.0890) | (0.0897) | (0.108) |
| O_CONCEN | −0.0632 | −0.0341 | −0.162 | −0.0309 | 0.00309 | −0.141 | −0.0309 | 0.00309 | −0.141 |
| | (0.0796) | (0.0780) | (0.105) | (0.0976) | (0.0984) | (0.119) | (0.0976) | (0.0984) | (0.119) |
| Constant | −0.00927 | −0.00384 | −0.0935 | −0.0234 | −0.0399 | −0.137 | −0.0234 | −0.0399 | −0.137 |
| | (0.0880) | (0.0850) | (0.114) | (0.102) | (0.103) | (0.124) | (0.102) | (0.103) | (0.124) |
| Wald chi$^2$/R$^2$ | 142.13 | 169.13 | 78.95 | 0.619 | 0.547 | 0.499 | 0.9229 | 0.9129 | 0.9162 |
| F-statistic | 142.13 *** | 169.13 *** | 78.95 *** | 8.000 *** | 5.960 *** | 4.910 *** | 120.00 *** | 89.34 *** | 73.59 *** |
| N | 90 | 90 | 90 | 90 | 90 | 90 | 90 | 90 | 90 |

Notes: * Significant at the 10% level. ** Significant at the 5% level. *** Significant at the 1% level.

Panel A of Table 6 indicates a significant positive relationship between board size and ROA, which means that board size enhances returns on bank assets. This result is consistent with theoretical perspectives of agency and resource dependency theories, and empirical evidence (e.g., Al-Farooque et al. 2020; Hamid and Purbawangsa 2022). This finding may highlight the advantage of the average number of board members suggested by the PDGB, which is around 10 members, as shown in the descriptive statistics in Table 5. However, this finding is inconsistent with the organizational behavior that proposes a negative association. On the other hand, the results reveal an insignificant influence of board size on ROE and Tobin's Q, which contradicts the theoretical perspectives. However, this finding is in line with the results of Y. A. Al-Matari (2022), Amrani et al. (2022), and Hamdan et al. (2019), who report an insignificant influence of board size on firm performance. The insignificant impact of board size on bank performance can be attributed to the significant influence of controlling shareholders and their representatives on boards of Saudi banks who have conflicting preferences and interests towards firm performance, which in turn weakens the role of size and independence of such boards. Thus, H1 is rejected.

For board composition, the results revealed an insignificant association between board independence and all measures of bank performance, which means that independent members on the board are unable to influence the bank performance. Despite this result being in line with some of the previous studies (e.g., Carter et al. 2010; Arosa et al. 2013; Roudaki 2018; Chatterjee and Nag 2022; Ararat and Yurtoglu 2021), it contradicted the

proposed positive relationship between the two variables, and thus H2 is also rejected. This result could be attributed to the nature of the ownership structure of the Saudi banking sector, where independent and non-executive directors may not be truly independent because controlling shareholders, such as family ownership and institutional ownership, dominate the Saudi-listed companies, and thus they have a strong influence on board composition with a tendency to assign board members with less independence to better serve their interests (Setia-Atmaja et al. 2009; Habtoor et al. 2019; Habtoor 2020).

Furthermore, the results revealed an insignificant association between executive members on the board and bank performance, which compels us to reject H3 as it contradicts the proposed significant influence of executive members on the board on bank performance. Empirically, this finding supports the result of Habtoor and Ahmad (2017), who find an insignificant impact of executive members on the board on corporate risk disclosure in Saudi non-financial listed companies. The insignificant role of executive members on the board of Saudi-listed banks is explainable by the marginal representation of executive members on the board, which hardly reaches 0.07 of board size, and some banks even have no executive members on their boards, as shown in the descriptive statistics in Table 5, which highlights banks compliance with the restrictions made by the PCGB on the maximum number of executive members on the board to mitigate their potential negative impacts on board effectiveness.

Regarding board education, this study categorized education into three categories with different measures. Board education (number of board members with at least a Bachelor's degree) has a significant negative impact on ROA and ROE, which implies that more educated members on the board (regardless of education level) lead to lower returns on assets and equity. However, board education had no impact on Tobin's Q. Therefore, H4 is rejected. Although, the results contradict the theoretical perspectives and most empirical evidence, they are consistent with the findings reported by Boadi and Osarfo (2019) and Fernanaez-Temprano and Tejerina-Gaite (2020). In this regard, Kaur and Singh (2019) argue that education may harm firm performance because of overconfidence biases of educated board members. In contrast, the regression results showed a significant positive relationship between both Master's and PhD holders on the board and bank performance. These results support the theoretical perspectives and previous empirical evidence (e.g., (Khidmat et al. 2020; Kim and Lim 2010) that the higher education of board members would lead to higher firm performance. Accordingly, H5 and H6 are accepted. This finding may encourage regulatory bodies and firms to take into account the educational level when appointing board members.

Board diversity is also categorized into gender diversity and nationality diversity. Gender diversity is not related to bank performance, as the regression results revealed an insignificant association between the presence of female members on the board and bank performance. This means that female members on Saudi banks' boards do not support bank performance, and thus H7 is rejected. This result is inconsistent with theoretical perspective that suggests a positive influence of a woman as a board member on firm performance. Nevertheless, this finding is in line with the result of Amrani et al. (2022), El-Chaarani et al. (2022), and Fernanaez-Temprano and Tejerina-Gaite (2020), who find no relationship between board gender and firm performance. This result can be explained by the fact that Saudi women are still underrepresented in the upper management of Saudi firms and their participation in board decisions is very limited. The descriptive statistics in Table 5 show a very low level of women representation of less than 0.09 members on the board (0.9% of board size), which limits their ability to influence board effectiveness and thus bank performance. This result highlights the marginalization of women on corporate boards (Boadi et al. 2022) and may indicate that the appointment of women is due to tokenism or is to fulfill part of their corporate social responsibility rather than evidence of a board's genuine intention to become gender diverse (Abdullah 2014).

On the other hand, and as expected, the results demonstrated a significant positive effect of CEO nationality on ROA and ROE, which indicates that CEO nationality is an

important factor for better operational and financial bank performance. Banks with non-Saudi or foreign CEOs achieve higher performance. By acquiring finer training, richer experience, and being more likely to be free from tribal and social relations and less subordination to family owners and other controlling shareholders, non-Saudi CEOs and board members are more likely to propose solutions of complex issues and have connections that facilitate the access to various resources to enhance bank performance. This result is consistent with the theoretical perspective that proposes a positive role of nationality and cultural diversity on firm performance. Moreover, this result is in line with the finding of Sarhan et al. (2019) who find a positive impact of board nationality on firm performance. On the other hand, CEO nationality does not affect Tobin's Q, and thus H8 is partly supported. These findings should be taken into account by the CMA and SAMA for further governance improvements by enhancing the poor role of women's participation in corporate boards and top management and encouraging firms for more national and cultural diversity.

With respect to board meeting attendance, the regression results showed a significant positive influence of the level of meeting attendance by board members on ROE. The results of the descriptive statistics in Table 5 indicated high levels of attendance (0.937), which may justify this result. However, an insignificant influence of board meeting attendance on ROA and Tobin's Q was found. These results bring partial support for agency and resource dependency theories and prior empirical evidence (e.g., Bhatt and Bhattacharya 2015; Chou et al. 2013; E. M. Al-Matari et al. 2022; AL Nasser 2020), and thus H9 is partly supported.

Board directorship does not affect bank performance, as the regression results demonstrated an insignificant relationship between the number of multiple directorships by board members and bank performance. Therefore, H10 is rejected, as the result contradicts the conflicted theoretical perspectives of resource dependency theory that suggests a positive influence of multiple directorships on firm performance, against the busyness hypothesis that proposes a negative association between the two variables. Nevertheless, this result is consistent with the findings of Y. A. Al-Matari (2022); Ferris et al. (2003); and Devos et al. (2009). Despite the descriptive statistics in Table 5 indicating that the average number of multiple directorships held by a board member is less than five memberships, there is a considerable number of directors who are also board members at the same time in a large number of non-listed companies, which can make them too busy and overworked. Therefore, such an issue should be of interest to the regulatory authorities.

Board IT experience was found to be significantly and positively associated with ROA and ROE. However, an insignificant impact of board IT experience on Tobin's Q was reported. This provides partial support for H11. This result indicates that hiring more board members with IT experience and knowledge improves bank operational and financial performance. This result is consistent with the theoretical argument of resource dependency theory, and it is in line with previous evidence (e.g., Zhang et al. 2018; Hamdan et al. 2019; Erkmen et al. 2020). This result highlights the important role of IT experience and awareness on Saudi firm performance in light of the kingdom's Vision 2030, which increasingly attracts international companies, together with the technology and advanced technical and managerial experience they have, which may put Saudi companies under competition pressure with foreign companies if they do not keep pace with developments in information technology. This finding may encourage Saudi regulatory authorities to regulate IT governance towards having corporate boards with experienced members in IT.

Regarding control variables, panel A of Table 6 indicates that the results were almost in line with theoretical perspectives and previous empirical evidence. Bank size was significantly and positively associated with ROA and Tobin's Q, whereas ROE was not affected. Bank age affected ROE and Tobin's Q significantly and negatively, while ROA was not influenced. Moreover, bank leverage was significantly and negatively associated with ROA and positively related to ROE and Tobin's Q. However, ownership concentration had an insignificant influence on all measures of bank performance.

*4.4. Robustness Check and Additional Analysis*

First, for comparison and robustness check of the FGLS regression results, the results of both the OLS and the random effects regression analysis were added, as shown in Panel B of Table 6.

Second, and as mentioned earlier, this study mainly relied on the FGLS estimator as the main method to analyze the data and test the hypotheses in the presence of heteroskedasticity and autocorrelation. However, Beck and Katz (1995) argue that this estimator may not be the ideal method to produce optimistic and robust standard error estimates when the panel's time dimension $T$ is smaller than its cross-sectional dimension $N$, which is the case in this study. Therefore, Beck and Katz (1995) suggest relying on the ordinary least square (OLS) with panel-corrected standard errors (PCSE). Accordingly, the FGLS models 1–3 in panel A of Table 6 were re-estimated using OLS coefficients with PCSE. Panel A of Table 7 reports the results of PCSE, which lend support to the earlier results reported in Panel A of Table 6 and provide new evidence as well. The significant positive impact of board size on ROA in Panel A of Table 6 became insignificant in Panel A of Table 7. Moreover, the insignificant negative influence of board independence on ROA and ROE in Panel A of Table 6 became significant in Panel A of Table 7, which means that board independent members on the board affect negatively the bank financial and operational performance. Despite this result being in line with some previous studies (e.g., Amin and Nor 2019; Kumar and Singh 2013; Waheed and Malik 2019; Dodd and Zheng 2022; Fariha et al. 2022), it contradicts the proposed positive role of board independence on bank performance. This result may highlight the negative role and influence of Saudi controlling shareholders on board composition with a tendency to assign members to the board with less independence to better serve their interests.

Third, endogeneity is a concern when it comes to investigating the association between corporate governance and firm performance (Wintoki et al. 2012; Giraldez-Puig and Berenguer 2018; Latif et al. 2020). The endogeneity problem can occur when board characteristics or explanatory variables and bank performance measures could be determined simultaneously or by unobservable factors (unobserved heterogeneity). In dealing with endogeneity, Li (2016) tests the prevailing econometric methods, such as instrumental variables, lagged dependent and independent variables, fixed effects, control variables, and GMM for dynamic models. The results indicated that all the prevailing econometric remedies work to mitigate the endogeneity bias to some degree. A large number of prior studies uses the instrumental variable technique (IV) to solve endogeneity (e.g., Dodd and Zheng 2022; Chen and Al-Najjar 2012; Kao et al. 2019). Following previous studies, this study checked for endogeneity through two steps: Firstly, the FGLS models 1–3 in panel A of Table 6 were re-estimated using the instrumental variables (IV) through the Two-Stage Least Squares (2SLS) regression, with the first-year lag of potential endogenous variables as instruments (e.g., Li 2016; Al-Farooque et al. 2020). To ensure the appropriateness of using the 2SLS, the instrumental variables should not be correlated with the error term of the models (Wooldridge 2002; Assenga et al. 2018; Wang et al. 2019). As shown in Table A1 of Appendix A, there was no correlation between the IVs and the error term.

Secondly, and before reporting the 2SLS regression results, it was important to determine whether it was necessary to use an IV approach instead of OLS. Therefore, the Durbin–Wu–Hausman test was run to determine if endogeneity was a problem in these models. The results of the Durbin–Wu–Hausman test indicated no endogeneity problem in the ROA and ROE models, as the results of the endogeneity test were insignificant (see Table 2), which means that the OLS method was the appropriate estimator for model 4 and 5 (Wooldridge 2002). However, the Durbin–Wu–Hausman test indicated an endogeneity problem in Tobin's Q model, as the result of the test was significant at the 0.01 level (see Table 2). Accordingly, the estimation of Tobin's Q model using the 2SLS approach provided more robust and reliable results than did OLS. Panel B of Table 7 reveals that the results of 2SLS for Tobin's Q were generally consistent with those reported in Panels A and B of Table 6. However, the insignificant positive relationship between board independence and

Tobin's Q became significant, indicating that independent board members enhance the market-based performance. This result is consistent with agency, resource dependency, and legitimacy theories perspectives, and it is in line with empirical evidence as well (e.g., (Dodd and Zheng 2022; Y. A. Al-Matari 2022; El-Chaarani et al. 2022). Thus, H2 is partly supported. Therefore, the conflicting role of board independence on bank performance needs to be carefully considered by the regulatory authorities, notably the CMA and SAMA, when developing further regulations.

**Table 7.** Results of additional analysis.

| Variables | Panel A: PCSE Model (1) ROA | Model (2) ROE | Model (3) Q | Panel B: 2SLS Model (4) Q |
|---|---|---|---|---|
| B_SIZE | 0.235 | 0.204 | 0.0262 | 0.128 |
| | (0.151) | (0.138) | (0.219) | (0.594) |
| B_IND | −0.135 ** | −0.123 * | 0.0768 | 0.565 ** |
| | (0.0654) | (0.0658) | (0.119) | (0.251) |
| B_EXE | 0.0240 | 0.00550 | −0.0730 | 0.171 |
| | (0.0925) | (0.0819) | (0.104) | (0.153) |
| B_EDU | −0.433 *** | −0.391 *** | 0.0132 | 0.102 |
| | (0.125) | (0.119) | (0.204) | (0.537) |
| B_MASTER | 0.236 *** | 0.200 *** | 0.326 *** | 0.418 *** |
| | (0.0795) | (0.0642) | (0.0949) | (0.146) |
| B_PHD | 0.255 *** | 0.253 *** | 0.0428 | −0.0710 |
| | (0.0909) | (0.0884) | (0.136) | (0.203) |
| GENDER | −0.232 | −0.217 | 0.0291 | 0.0132 |
| | (0.286) | (0.306) | (0.335) | (0.433) |
| CEO_NAT | 0.272 *** | 0.331 *** | −0.0498 | −0.239 |
| | (0.0862) | (0.0909) | (0.118) | (0.217) |
| B_ATTEND | 0.0823 * | 0.154 *** | −0.00999 | −0.00967 |
| | (0.0483) | (0.0482) | (0.0625) | (0.106) |
| B_DIRECT | 0.0577 | 0.0842 | −0.00187 | 0.102 |
| | (0.0891) | (0.0935) | (0.161) | (0.126) |
| B_IT_EXP | 0.267 ** | 0.207 ** | 0.00486 | −0.0307 |
| | (0.107) | (0.102) | (0.140) | (0.227) |
| F_SIZE | 0.247 *** | 0.173 *** | 0.229 ** | 0.204 |
| | (0.0670) | (0.0535) | (0.0985) | (0.191) |
| F_AGE | −0.187 *** | −0.227 *** | −0.406 *** | −0.312 ** |
| | (0.0671) | (0.0656) | (0.112) | (0.153) |
| F_LEV | −0.292 *** | 0.242 ** | 0.305 ** | 0.480 *** |
| | (0.104) | (0.103) | (0.143) | (0.151) |
| O_CONCEN | 0.0366 | 0.0550 | −0.104 | −0.139 |
| | (0.0814) | (0.0850) | (0.0752) | (0.155) |
| Constant | −0.00406 | −0.0247 | −0.106 | −0.148 |
| | (0.115) | (0.108) | (0.201) | (0.180) |
| Wald chi$^2$/Adj. R$^2$ | 0.522 | 0.4642 | 0.3774 | 0.365 |
| F-statistic | 849.01 *** | 620.52 *** | 590.86 *** | 77.24 *** |
| N | 90 | 90 | 90 | 87 |

Notes: * Significant at the 10% level. ** Significant at the 5% level. *** Significant at the 1% level.

## 5. Conclusions, Limitations, and Future Research

The purpose of this study is to investigate the impact of board characteristics on Saudi bank performance, in light of Saudi corporate governance regulations.

The study employs the FGLS as the main method to analyze the data set of all 12 Saudi listed banks over a period of ten years from 2009 to 2018. However, the OLS and random effects models are used for comparison and robustness check purposes, while the PCSE and 2SLS models are conducted for additional analysis.

The empirical evidence largely supports the study arguments. The results indicate that among the board characteristics investigated, board size, composition, education, nationality diversity, IT experience, and attendance are significant determinants of Saudi bank performance. The results of this study have several theoretical and practical implications. First, the current study relies on multiple theoretical perspectives drawn from several theories for a deeper understanding of how and which board characteristics could affect bank performance. Second, this study extends the existence literature on corporate

governance by providing empirical evidence from Saudi Arabia on the influence of various characteristics of the board on bank performance. Third, the empirical results have important implications for Saudi policymakers and regulatory authorities, including the CMA and SAMA, to assess the current practice of governance in the banking industry, and to construct an appropriate set of governance mechanisms. Companies and market participants might use these findings to shape their understandings of the role of boards of directors on bank performance.

This study is subject to some limitations that could be potential avenues for future research. First, based on the availability of data, the current study investigates a set of board characteristics on the performance of Saudi listed banks. However, more board characteristics and different measures of bank performance may be added, and non-listed banks could be included to overcome the disadvantages of the small sample, strengthen the results, and gain a deeper understanding of the role of board characteristics on bank performance. Furthermore, comparative studies among GCC countries will be of value for a deeper understanding of the determinants of bank performance in these countries that share similar cultural values and institutional systems. Second, while this study focuses on board characteristics, further research may look more deeply into more and different potential drivers of bank performance, such as the chairperson and CEO attributes, and board committees' characteristics. Third, as this study attributes the poor role of board composition on bank performance to the potential significant influence of concentration ownership in Saudi Arabia, investigating the potential moderating role of different types of ownership and controlling shareholders, such as family, government, institutional, and foreign ownership, on the relationship between board composition and bank performance is essential.

**Funding:** This research was funded by Northern Border University, Grant number CMR-2019-1-10-F-1112, and the APC was funded by CMR-2019-1-10-F-1112.

**Institutional Review Board Statement:** Not applicable.

**Informed Consent Statement:** Not applicable.

**Data Availability Statement:** The data presented in this study are available upon request from the author.

**Conflicts of Interest:** The author declares no conflict of interest.

# Appendix A

**Table A1.** Correlation between instrumental variables and error terms of the study models.

| Variables | (1) | (2) | (3) | (4) | (5) | (6) | (7) | (8) | (9) | (10) | (11) | (12) | (13) | (14) | (15) | (16) | (17) | (18) | (19) | (20) | (21) |
|---|---|---|---|---|---|---|---|---|---|---|---|---|---|---|---|---|---|---|---|---|---|
| (1) ROA | 1.000 | | | | | | | | | | | | | | | | | | | | |
| (2) ROE | 0.756 ** | 1.000 | | | | | | | | | | | | | | | | | | | |
| (3) Q | 0.423 ** | 0.416 ** | 1.000 | | | | | | | | | | | | | | | | | | |
| (4) B_SIZE_lag1 | 0.340 ** | 0.314 ** | 0.378 ** | 1.000 | | | | | | | | | | | | | | | | | |
| (5) B_IND_lag1 | 0.015 | −0.123 | 0.208 * | 0.019 | 1.000 | | | | | | | | | | | | | | | | |
| (6) B_EXE_lag1 | 0.116 | 0.123 | 0.006 | −0.223 * | −0.259 * | 1.000 | | | | | | | | | | | | | | | |
| (7) B_EDU_lag1 | 0.528 ** | 0.342 ** | 0.244 * | 0.423 ** | −0.117 | 0.176 | 1.000 | | | | | | | | | | | | | | |
| (8) B_MASTER_lag1 | −0.253 | −0.384 ** | −0.176 | −0.139 | 0.247 * | −0.049 | −0.113 | 1.000 | | | | | | | | | | | | | |
| (9) B_PHD_lag1 | 0.131 | 0.067 | 0.170 | 0.745 ** | −0.125 | −0.246 * | 0.366 ** | 0.138 | 1.000 | | | | | | | | | | | | |
| (10) GENDER_lag1 | −0.086 | 0.092 | −0.037 | 0.046 | −0.140 | 0.043 | 0.202 | 0.072 | −0.018 | 1.000 | | | | | | | | | | | |
| (11) CEO_NAT_lag1 | 0.348 ** | 0.412 ** | 0.089 | 0.314 ** | −0.060 | 0.333 ** | 0.420 ** | −0.297 ** | 0.063 | 0.323 ** | 1.000 | | | | | | | | | | |
| (12) B_ATTEND_lag1 | 0.119 | 0.063 | −0.139 | 0.105 | −0.093 | −0.088 | 0.212 * | 0.074 | 0.336 ** | −0.056 | −0.052 | 1.000 | | | | | | | | | |
| (13) B_DIRECT_lag1 | 0.175 | 0.284 ** | 0.259 * | 0.284 ** | −0.311 ** | −0.192 | 0.095 | −0.392 ** | 0.362 ** | −0.115 | 0.028 | 0.131 | 1.000 | | | | | | | | |
| (14) B_IT_EXPlag1 | 0.215 * | 0.076 | −0.007 | −0.136 | −0.055 | 0.075 | −0.098 | −0.422 ** | −0.299 ** | −0.150 | −0.199 | −0.113 | −0.145 | 1.000 | | | | | | | |
| (15) F_SIZE_lag1 | 0.458 ** | 0.317 ** | 0.248 * | 0.427 ** | −0.083 | −0.026 | 0.271 * | −0.286 * | 0.179 | −0.294 ** | 0.116 | 0.170 | 0.382 ** | 0.148 | 1.000 | | | | | | |
| (16) F_AGE_lag1 | −0.139 | −0.031 | −0.408 ** | −0.183 | −0.043 | −0.247 * | −0.122 | −0.018 | 0.184 | 0.042 | 0.301 ** | 0.055 | 0.086 | −0.102 | 0.332 ** | 1.000 | | | | | |
| (17) F_LEV_lag1 | −0.028 | 0.463 ** | 0.081 | 0.022 | −0.316 ** | 0.095 | −0.101 | −0.369 ** | −0.071 | 0.292 ** | 0.202 | 0.043 | 0.247 * | −0.027 | −0.141 | 0.332 ** | 1.000 | | | | |
| (18) O_CONCEN_lag1 | 0.192 | 0.194 | −0.027 | 0.254 * | −0.076 | −0.155 | 0.537 ** | 0.042 | 0.323 ** | 0.379 ** | 0.333 ** | 0.334 ** | 0.136 | −0.206 * | 0.211 | 0.385 ** | 0.150 | 1.000 | | | |
| (19) Error_Term_ROA | 0.618 ** | 0.639 ** | 0.327 ** | −0.002 | 0.116 | −0.047 | 0.020 | −0.064 | −0.036 | −0.011 | 0.006 | −0.125 | 0.108 | 0.113 | −0.118 | −0.004 | 0.039 | −0.097 | 1.000 | | |
| (20) Error_Term_ROE | 0.586 ** | 0.673 ** | 0.267 * | −0.009 | 0.096 | −0.021 | 0.008 | −0.032 | −0.041 | −0.021 | 0.004 | −0.132 | 0.099 | 0.082 | −0.082 | 0.002 | 0.076 | −0.108 | 0.949 ** | 1.000 | |
| (21) Error_Term_Q | 0.286 ** | 0.254 * | 0.708 ** | 0.030 | 0.303 | −0.055 | −0.018 | −0.055 | 0.015 | −0.014 | −0.027 | −0.174 | 0.186 | 0.045 | −0.022 | 0.010 | −0.084 | −0.037 | 0.462 ** | 0.377 ** | 1.000 |

ROA is return on assets; ROE is return on equity; Q is Tobin's Q; B_SIZE_lag1 is the one year lagged values of board size; B_IND_lag1 is the one year lagged values of board independence; B_EXE_lag1 is the one year lagged values of executive members on the board; B_EDU_lag1 is the one year lagged values of educated board members; B_MASTER_lag1 is the one year lagged values of board members with a Master's degree; B_PHD_lag1 is the one year lagged values of board members with a PhD degree; GENDER_lag1 is the one year lagged values of gender diversity; CEO_NAT_lag1 is the one year lagged values of CEO nationality diversity; B_ATTEND_lag1 is the one year lagged values of board meeting attendance; B_DIRECT_lag1 is the one year lagged values of multiple directorships of board members; B_IT_EXP_lag1 is the one year lagged values of board members with IT experience; F_SIZE_lag1 is the one year lagged values of bank size; F_AGE_lag1 is the one year lagged values of bank age; F_LEV_lag1 is the one year lagged values of bank leverage; O_CONCEN_lag1 is the one year lagged values of ownership concentration; Error_Term_ROA is the error term of ROA model; Error_Term_ROE is the error term of ROE model; Error_Term_Q is the error term of Tobin's Q model. Notes: * Significant at the 5% level. ** Significant at the 1% level.

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
