# Peer review of "Board Attributes and Bank Performance in Light of Saudi Corporate Governance Regulations"

_jrfm, doi:10.3390/jrfm15100441_

Round 1

Reviewer 1 Report

Originality: There is a need to revise the paper title, and the current title does not portray the actual meaning of the paper. 

Abstract: The abstract is not well written. There is a need to revise with explicit contents of the abstract, i.e., the main issue, sampling, a statistical tool, methods, results, and implication. The author(s) should provide a precise and focused abstract

  • The sampling criteria, population, year, and unit of analysis for selecting companies are missing. The author should highlight sampling criteria for more clarity to readers.
  • As a suggestion for improvement, the author(s) should not use the same Keywords as like Paper Title. It is encouraged to used different keywords which are not in the Paper title. It will enhance paper searchability after publication. 

Introduction:

      The introduction section is not well written. There are ambiguous statements and no clarity in the introduction section.

    • The introduction section is not started with a broader area and issue or in a global context. Therefore, there is no synthesis in writing an introduction section.
  • Relationship to Literature: The paper incorporated major literature of corporate governance containing among  companies, but the paper does not sufficiently cover recent research in the area. Helpful in this regard would be to include relevant research recently covered in top journals of similar scope. Further, work needs to be done to support the findings based on the current literature, as a recent theory in the area is directly counter to what was found.

    • There is a need to add more critical recent literature and based on theoretical argumentation.
    • he hypotheses development is poorly written; author(s) should cite previous studies relevant to proposed hypotheses, i.e., international and local perspectives studies in the light of underpinning theory.
    • Conclusion: Author(s) should provide concluding statements rather than repetitive statements in the conclusion portion.

               It is highly recommended to write the conclusion section separately from the discussion of the findings

    • The reviewer found that author(s) has cited only few  recently published papers in this article (Most of the cited articles are ten years old). As a suggestion, the author(s) must cite new articles (latest literature) to make holistic discussion and sturdy paper with high readability.

Reviewer 2 Report

I would like to congratulate the author(s) on this work. The paper is well grounded theoretically based on the postulates of one of the most important divulgation theories: agency theory and others as well. The paper is adequately developed in what concerns methodology and research method. Moreover, there is an adequate link between all the parts of the paper. The findings are informative and potentially very useful in the studied context.

One minor suggestion which I must hare that author may add one more hypothesis that the critical mass of gender is more pronounced in the said relation rather than the token participation. in doing so author may add one hypothesis after H7. 
